# Co-Channel Compatibility Analysis of IMT Networks and Digital Terrestrial Television Broadcasting in the Frequency Range 470–694 MHz Based on Monte Carlo Simulation

**DOI:** 10.3390/s23218714

**Published:** 2023-10-25

**Authors:** Hussein Taha, Péter Vári, Szilvia Nagy

**Affiliations:** 1Doctoral School of Multidisciplinary Engineering Sciences, Széchenyi István University, 9026 Győr, Hungary; taha.hussein@sze.hu; 2Department of Telecommunications, Széchenyi István University, 9026 Győr, Hungary; nagysz@sze.hu

**Keywords:** IMT, DTTB, broadcasting, coexistence, 470–694 MHz, WRC-23, digital dividends, interference, Monte Carlo simulation, SEAMCAT

## Abstract

According to Resolution 235 adopted during the World Radiocommunication Conference 2015 (WRC-15) and the preliminary agenda item 1.5 of WRC-2023, the International Telecommunication Union Radiocommunication Sector (ITU-R) has been entrusted to carry out sharing and compatibility studies between broadcasting and mobile services in the frequency range 470–694 MHz in Region 1. This study specifically focuses on evaluating electromagnetic compatibility in potential co-channel sharing scenarios between Digital Terrestrial Television Broadcasting (DTTB) and International Mobile Telecommunications (IMT) systems within the 470–694 MHz frequency band that may arise in the foreseeable future. To assess the conditions for coexistence, a Monte Carlo simulation method implemented through the SEAMCAT software version 5.4.2 is employed, examining six potential interference scenarios. The simulation results yield the minimum coordination distance required between IMT and DTTB services in the 470–694 MHz band based on the protection criteria to ensure harmonious coexistence while maintaining satisfactory performance. Furthermore, the study investigates the impact of various factors such as transmitter power, antenna heights, coverage radius, antenna discrimination, and antenna tilt angle on the separation distances. The focus lies primarily on critical interference scenarios across neighboring countries’ borders. The simulation outcomes confirm that sharing the same frequency band between IMT and DTTB networks would result in significant mutual interference. Nevertheless, carefully analyzing diverse parameters and assumptions helped provide recommendations to reduce the required separation distances. These findings are valuable for broadcasters, mobile operators, and regulators in establishing the technical coexistence conditions for DTTB and IMT in the new band.

## 1. Introduction

The International Telecommunication Union Radiocommunication Sector (ITU-R) plays a crucial role in allocating and managing radio spectrum resources, promoting the advancement of digital services such as television broadcasting and mobile communications [1]. 

With the transition from analog to digital broadcasting, a significant portion of the radio frequency spectrum has become available for other services, commonly known as the “digital dividend” [2]. The frequency allocations of digital dividend bands are collectively decided at ITU World Radiocommunication Conferences (WRCs). Figure 1 illustrates the historical approach following WRCs to allocate the radio spectrum among various over-the-air services in ITU Region 1.

The first digital dividend band (from 790 to 862 MHz) was allocated following the WRC-07 in 2007, which identified it as suitable for International Mobile Telecommunications (IMT) services [3]. After the success of the first digital allocation band, the ITU-R allocated a second digital dividend band (from 694 to 790 MHz) to mobile services following WRC-12 in 2012 [4,5,6].

Driven by the escalating demand for mobile broadband services and the development of new technologies, there is growing pressure to allocate new digital dividends in the 470–694 MHz band to accommodate mobile services. This prompted the ITU to undertake extensive consultations and discussions with relevant stakeholders during subsequent WRCs on the possibility of granting a new allocation for IMT services.

In this regard, Resolution 235 was adopted at WRC-15, calling for thorough investigations on sharing and compatibility between broadcasting and mobile services in the frequency range 470–960 MHz in Region 1 [5,6]. Furthermore, the preliminary agenda item 1.5 of the upcoming WRC-23 calls for an assessment of the spectrum requirements for existing services in the 470–694 MHz band in Region 1 and explores the possibility of granting a new allocation for IMT services in all or parts of the band [7]. 

The frequency range 470–694 MHz is primarily used for DTTB services. DTTB provides consumers access to a range of free-to-air television channels and is a key means of distributing television content in many countries worldwide. Additionally, this band accommodates various other services, including radio microphones and in-ear monitoring systems, which play a crucial role in the performing arts and broadcasting industries. These services require access to high-quality spectrum resources to operate effectively and reliably. Consequently, granting a new allocation for IMT in the 470–694 MHz frequency band will raise concerns regarding potential interference with existing broadcasting services. 

In light of these concerns, our recent article [8] delves into an in-depth exploration and analysis of potential options for the future utilization of the 470–694 MHz band in Europe. The article extensively discussed the benefits and implications of each option, recognizing the need for a flexible strategy that can adapt to the evolving realities of both the broadcasting and mobile markets.

The availability of studies specifically addressing the coexistence of IMT and broadcasting services in the 470–694 MHz frequency range remains limited. Consequently, studies conducted on previous digital dividend bands have been referenced to gain insights into the sharing and compatibility aspects between these services.

Different approaches have been employed in the previous literature to investigate coexistence issues, including link budget analysis, simulations, laboratory measurements, and field measurements. Each approach serves a distinct purpose and has advantages and limitations. Our paper in [9] overviews these coexistence approaches and reviews the relevant literature for each method. To facilitate a clear understanding, Table 1 compares the purpose, advantages, and limitations of coexistence approaches.

These approaches complement each other in investigating coexistence issues. Link budget analysis provides a theoretical foundation, simulations offer flexibility and scenario testing, laboratory measurements allow for controlled testing, and field measurements provide real-world insights. Typically, these methods are combined to thoroughly evaluate and ensure coexistence between DTTB and IMT systems in digital dividend bands. The choice of methodology depends on the specific research objectives and available resources.

The paper [10] conducts a simulation study using terrain-aware simulation software to assess the coexistence of IMT and DTTB systems in the 470–694 MHz band, focusing on locations in Saudi Arabia. The research quantifies interference levels and protection ratios, finding that reasonable separation distances between networks are possible in most cases. However, the impact of DTTB on IMT uplink is challenging, requiring additional mitigation methods. The required separation distance varies by terrain, with mountains needing the least and sea borders requiring more separation. Mitigation techniques include frequency reallocation and turning off IMT sectors facing DTTB networks to reduce separation distances.

Refs. [11,12,13,14] used Monte Carlo simulations by means of the Spectrum Engineering Advanced Monte Carlo Analysis Tool (SEAMCAT) and the Minimum Coupling Loss (MCL) method to analyze interference. They examined the compatibility between 4G/5G and DTTB systems in the 700 MHz band, particularly in scenarios involving neighboring countries. The results highlighted the need for significant separation distances to ensure adequate system performance.

The authors of Refs. [15,16] validated the SEAMCAT simulation tool by comparing its output to actual measurements from their experiments, concluding that SEAMCAT is valuable for simulating complex scenarios. 

Ref. [17] used simulations to assess LTE interference on DTTB services in collective antenna systems with wideband amplifiers. Ref. [18] found that inserting a 1 MHz guard band between LTE and DTTB reduced interference significantly. Ref. [19] showed that LTE and DTTB systems can coexist in the 700 MHz band while maintaining protective distances. The study [20] calculated DTTB interference on IMT channels in the regions bordering Italy, indicating the need for over 100 km of separation to protect IMT systems from DTTB interference.

In ref. [21], solutions to mitigate interference from DTTB in LTE systems included reducing DTTB transmission power and maintaining a minimum distance between LTE base stations and DTTB. Ref. [22] suggested co-locating TV transmitters and cellular base stations on the same tower to ensure coexistence in the 700 MHz and 800 MHz bands, offering consistent protection ratios and cost savings. However, this approach may pose challenges for low-power TV transmitters due to strict separation distance requirements. Monte Carlo simulations in [23] emphasized the need for guard bands and minimum separation distances to ensure electromagnetic compatibility between DTTB and 5G networks in the 700 MHz band. The study [24] focused on determining protection ratios to protect primary DTTB users from other devices utilizing TV white spaces.

The articles in [25,26] focus on the coexistence of 5G networks with other services in the 6425–7125 MHz band. They employ Monte Carlo simulation to address interference issues. In [25], the study identified protection distances and frequency offsets needed for 5G and fixed services to coexist compatibly. Different protection criteria, like I/N and C/I, were used to determine suitable separation distances, especially in co-location scenarios within a single administration. In [26], the research evaluated aggregate interference from 5G base stations within satellite footprints, considering factors like signal-to-noise degradation and bit error rates for fixed-satellite service (FSS) bent-pipe transponders. The results suggested that coexistence between 5G NR and satellite systems in this frequency range is feasible, with no significant adverse effects on satellite link performance.

Refs. [27,28] assessed the coexistence of DTTB and LTE networks in the 700 MHz and 800 MHz bands, using laboratory measurements and link budget analysis considering LTE signal factors. They highlighted significant challenges in mitigating interference. Ref. [29] studied the impact of DTTB on LTE downlink in the 800 MHz band through laboratory measurements, considering various factors affecting LTE performance. Laboratory experiments in refs. [30,31,32,33] confirmed the effectiveness of low-pass filters in reducing IMT interference in digital TV channels. Ref. [34] found negligible LTE downlink interference on different DTTB setups in the 800 MHz band, especially for portable indoor reception. Ref. [35] conducted laboratory measurements to investigate LTE uplink interference on coaxial cables connecting TV sets to digital TV aerial sockets. Ref. [36] explored TV white space utilization by indoor LTE-A femtocells with rooftop DTTB reception, validated in laboratory and real-world settings. Ref. [37] used laboratory experiments to determine protection ratios and overload thresholds between analog and digital terrestrial TV systems and LTE in the 700 MHz band.

The article cited in [38] provides a comprehensive guide on conducting field measurements to study the coexistence of DTTB and IMT in the UHF broadcasting band. Field measurements were integral to several earlier research studies [15,24,30,36], supplementing simulations and laboratory observations. In [15], field measurements validated outcomes from the SEAMCAT simulation tool. Ref. [24] focused on establishing the range of protection ratios when other devices utilized TV white spaces. Ref. [30] confirmed the real-world effectiveness of an RF band stop filter in mitigating LTE interference in DTTB reception. In ref. [36], field measurements were conducted to verify the feasibility of TV white space utilization for indoor LTE-A femtocells in rooftop DTTB reception setups. In ref. [39], interference scenarios were assessed following ITU guidelines, revealing significant detrimental effects of LTE UE signals on indoor DTTB reception and decreased uplink data rates when assessing DTTB interference with LTE uplink. The paper cited in [40] presented an indoor short-range distribution system for wireless retransmission of DTTB-compliant content in free TV channels. The system utilized geo-location databases, spectrum sensing, and power control techniques to protect existing services. The paper included computer simulations, real-world testing, and the development of an algorithm for channel selection. The results showed promising coverage for typical home environments without causing interference to nearby receivers.

In wireless communication systems, it is important to emphasize the significance of index modulation and spatial modulation technologies as part of a broader strategy to enhance co-channel compatibility [41,42,43]. Index modulation improves coexistence by dividing the frequency spectrum into subchannels and selectively applying index modulation to certain subchannels based on factors like channel quality and interference levels. Dynamic adaptation mechanisms can continuously monitor channel conditions and adjust modulation schemes in real time, enabling seamless switching between subchannels using index modulation and conventional schemes. When multiple users share a sub-channel and use index modulation, ensuring their index values are orthogonal enhances coexistence. However, implementing index modulation in the co-channel operation between IMT and DTTB systems requires careful design, coordination, and adaptation mechanisms, depending on specific system requirements and available resources.

The existing literature primarily focuses on specific scenarios and use cases, whereas our present contribution comprehensively examines all potential coexistence scenarios in the 470–694 MHz band. 

Monte Carlo simulation by means of the SEAMCAT software version 5.4.2 is the method adopted for our study, due to several compelling reasons. It offers a cost-effective, safe, controlled, repeatable, and flexible way to assess system interactions, accelerating the analysis process and providing valuable insights for decision-making and system design. Simulations grant complete control over test scenarios, allowing the creation of a wide range, including worst-case scenarios, which may be difficult or impossible to conduct in the real world. This control facilitates comprehensive testing and analysis. Additionally, international spectrum regulatory authorities follow the practice of simulating coexistence scenarios between DTTB and IMT systems, as specified in international agreements.

This contribution determines the coordination distances necessary for deploying IMT networks in the presence of existing broadcasting services in co-channel scenarios, considering both rural and urban areas. Moreover, the study evaluates the combined impact of various factors, such as transmission power, coverage area, antenna height, antenna discrimination, and antenna tilt angle, on the required separation distances in the most common interference scenarios between DTTB and IMT networks deployed in neighboring countries. These simulation results can help the relevant stakeholders make informed decisions and implement appropriate technical and organizational measures to ensure the effective and efficient operation of DTTB and IMT services in the 470–694 MHz band.

The remainder of the paper is structured as follows. Section 2 presents the possible coexistence scenarios. Section 3 is devoted to simulation parameters, propagation models, and protection criteria. Section 4 describes the methodology followed for each coexistence scenario. Section 5 is devoted to simulation results and compatibility analysis. Finally, Section 6 concludes the article and summarizes the key findings.

## 2. Coexistence Scenarios for DTTB and IMT

The coexistence of IMT networks and DTTB in the 470–694 MHz band can give rise to six possible interference scenarios. These scenarios are classified based on the potential sources of interference (interferer) and the affected entities (victim). The interferer can be an IMT base station (IMT-BS), IMT user equipment (IMT-UE), or a DTTB transmitter. The victim can be a fixed rooftop DTTB reception, a portable DTTB indoor reception, an IMT-BS, or an IMT-UE. Figure 2 depicts the potential coexistence scenarios that may occur when IMT and DTTB systems coexist in the 470–694 MHz band.

Among the six possible interference scenarios, Scenario (A) involves an IMT-BS interfering with a fixed outdoor DTTB receiver. This scenario is particularly worse when the IMT-BS is situated in close proximity to the DTTB rooftop antenna and pointed in the same direction as the TV station’s transmission. In Scenario (B), an IMT-BS interferes with a portable DTTB indoor receiver, although the impact is less critical due to the additional signal loss caused by building penetration. Scenario (C) involves an IMT-UE interfering with a fixed rooftop DTTB receiver, with maximum interference when the IMT-UE is outdoors and near the DTTB antenna. In Scenario (D), an IMT-UE interferes with a portable DTTB indoor receiver, with the worst case being both devices in the same room.

The worst case for scenarios (A), (B), (C), and (D) involves the DTTB receiver being positioned at the coverage edge, receiving the minimum required DTTB signal power, while the IMT-BS or IMT-UE transmits at the maximum power level.

Scenario (E) occurs when a DTTB transmitter interferes with an IMT-BS, and Scenario (F) involves a DTTB transmitter interfering with an IMT-UE. The most critical circumstances for both (E) and (F) arise when the DTTB transmitter is a high-tower high-power (HTHP) class, transmitting at the maximum power level.

Scenarios (A) and (E) are particularly significant for IMT and DTTB coexistence in the 470–694 MHz band, especially near border regions where neighboring countries may have different service deployments within the same frequency range. These scenarios also apply to situations within a single country and frequency band.

## 3. Technical Characteristics

### 3.1. IMT Parameters

The general technical parameters of the IMT system are based on the contribution 28 supplied by Working Party 5D to ITU Task Group TG 6/1 [44]. Table 2 and Table 3 summarize the technical parameters of IMT systems for frequency bands below 1 GHz.

### 3.2. DTTB Parameters

The technical specifications and operational parameters of the DTTB system in the 470–694 MHz band are based on Report ITU-R BT.2383-4 and the contribution 32 supplied by Working Party 6A to Task Group 6/1 [45,46]. Table 4 summarizes the characteristics and parameters of the DTTB system in the 470–694 MHz band.

### 3.3. Propagation Models

ITU-R Recommendation P.1546-6 is commonly employed for interference path calculation in sharing and compatibility studies [47]. This propagation model enables the prediction of point-to-area radio propagation for terrestrial services. It considers different types of environments, such as urban, suburban, and rural areas, and provides predictions for land, warm sea, and cold sea paths. In the context of our study, only the land path with different time percentages for rural and urban environments was considered. Additionally, Recommendation ITU-R P.2109-1 was also used when the path includes indoor UE [48]. 

### 3.4. Protection Criteria

The basic protection criteria, as stated in refs. [49,50], specify that the total interference generated by new sources should not exceed a certain fixed percentage of the overall noise power of the system. This criterion serves to ensure that introducing new services or technologies, such as IMT in the 470–694 MHz band, does not cause excessive interference that would degrade the performance of existing services, such as DTTB. Adhering to the basic protection criteria helps balance accommodating new services and technologies while safeguarding the operation of existing services, ultimately ensuring harmonious coexistence and effective spectrum management.

The interference-to-noise ratio for each simulation case can be calculated in the general format:$$I/N=P_{t}+G_{t}(\theta_{t})+G_{r}(\theta_{r})-PL-OL-N$$
where:
I/N: Interference-to-noise ratio, dBm/MHz$P_{t}$: Transmitter power density, dBm/MHz $G_{t}(\theta_{t})$: Transmitter antenna gain, dBi $G_{r}(\theta_{r})$: Receiver antenna gain, dBi PL: Propagation loss, dB OL: Other loss, like feeder loss or body loss, dB N: Noise floor, dBm/MHz.

In the case of DTTB services as the victim, the interference-to-noise ratio (I/N) is compared with the protection criterion of I/N = −10 dB, as specified in Recommendation ITU-R BT.1895 and for the protection of DTTB from other co-primary services [49]. This criterion ensures that the interference experienced by DTTB receivers remains below a certain threshold relative to the background noise, thus preserving the quality and reliability of the DTTB service. Additionally, according to Report ITU-R BT.2383 and GE06, 95% of the locations in the small area at the edge of coverage should be protected, corresponding to an interference probability of less than 5% [2,14].

For IMT networks as a victim, the result of interfering power-to-noise power ratio is compared to the protection criterion of I/N = −6 dB, as given in Report ITU-R M.2292-0 [50] and referred to in Section 2.9 of Annex 1 of Recommendation ITU-R M.2101 [51]. This criterion ensures that the interference received by IMT systems remains below a certain threshold relative to the background noise level, providing adequate protection to maintain the desired performance of the IMT network. Additionally, to ensure sufficient protection, the average bitrate loss at the IMT receiver should not exceed 5%, as stated in Recommendation 3GPP TS 36.104 [52].

These protection criteria play a crucial role in assessing the coexistence of IMT and DTTB systems in the 470–694 MHz band. By comparing the interference levels with these criteria, regulators, operators, and other stakeholders can determine the required separation distances, transmission power limits, or other technical measures necessary to ensure the desired level of protection for both services.

## 4. Methodology

Monte Carlo simulation is one of the approaches used for electromagnetic compatibility analysis and in-depth assessment of interference probabilities between complex systems under different conditions. One software tool commonly used for the Monte Carlo simulation of radiocommunication systems is SEAMCAT software version 5.4.2 (Spectrum Engineering Advanced Monte Carlo Analysis Tool). SEAMCAT allows users to model a wide range of radiocommunication systems, including DTTB and IMT services, and to evaluate the effects of various scenarios on system performance.

This study uses a Monte Carlo simulation by means of the SEAMCAT software version 5.4.2, described in Report ITU-R SM.2028, to analyze the co-channel compatibility between DTTB and IMT-2020 services in the 470–694 MHz band for urban and rural areas [53]. The simulation employs the technical parameters summarized in the aforementioned tables to assess the level of interference and determine the minimum separation distance for compatibility based on the protection criteria for all coexistence scenarios mentioned in Section 2. Furthermore, extensive investigations are carried out on the most common interference scenarios when DTTB and IMT networks are deployed in neighboring countries: Scenario A and Scenario E. 

For IMT network planning, the IMT network model consists of 7 base stations (BS) arranged in a hexagonal pattern with 3 sectors. The cell radius is set to 1.5 km for urban scenarios and 8 km for rural scenarios. Each sector serves 3 users, uniformly deployed within the base station region based on the cell radius. For each given distance, around 10,000 positions are randomly selected, and 10,000 frames are transmitted from each IMT BS.

Regarding the broadcasting planning for DTTB, the focus is on attaining a precise probability of reception at a specific location within a limited area commonly referred to as a pixel, which typically measures 100 m × 100 m [45,46]. This probability represents the percentage of locations where the DTTB receiver operates flawlessly within a given timeframe.

The coordination distance is calculated from the center of the IMT network to the DTTB service. To ensure the reliability of the results, an extensive number of samples/events (10,000) are simulated and analyzed.

The evaluation methodology followed in this article consists of two main parts.

### 4.1. Interferences from the IMT Systems to the DTTB

#### 4.1.1. Scenario A

IMT Base stations interfere with fixed outdoor DTTB receivers in rural and urban areas, as shown in Figure 2A.

Monte Carlo simulations are utilized to assess the level of interference between the IMT network and the outdoor fixed DTTB receiver, aiming to determine the minimum coordination distances while adhering to the protection criterion of I/N = −10 dB. These simulations also ensure that 95% of the locations within the small coverage area receive the necessary protection. The analysis considers four different propagation model times (1.75%, 5%, 10%, and 50%) across a 100% land path. The 1.75% time is specifically evaluated using Recommendation ITU-R P.1546-6, following the simplified method outlined by Study Group 3 in their liaison to TG 6/1 (Doc 6-1/31) [47,54].

Within this scenario, two different load factors (LFs) of the IMT network are implemented: 50% LF and 20% LF. The variation in the full base station equivalent isotopically radiated power (BS EIRP) associated with the network loading factor is carefully analyzed at the prescribed separation distance [51]. The mean value for BS EIRP is determined to be 44.3 dBm, considering the Gaussian distribution used to simulate the power variation of the IMT base station over time, while a value of 58 dBm is also evaluated, aligning with the baseline parameters [44,52].

Furthermore, the study investigates the impact of increasing the IMT-BS antenna tilt angle (ranging from 0 to −7 degrees) on the probability of interference in the DTTB receiver. 

The simulations also account for variations in TV transmitter power values and TV coverage areas, corresponding to different classes of DTTB transmitters such as HTHP transmitters, −6 dB reduced power HTHP transmitters, and MTMP transmitters [45].

In the context of IMT network interference with a DTTB receiver, two model configurations are considered: one with full DTTB antenna discrimination and the other with no DTTB antenna discrimination [45,46].

In full DTTB antenna discrimination, the DTTB receiver is equipped with advanced features and capabilities to distinguish between desired DTTB signals and unwanted interference from the IMT network. The receiver uses multiple antennas or advanced signal processing techniques to identify and prioritize DTTB signals over the interference caused by the IMT network, ensuring high-quality DTTB reception. 

In contrast, no DTTB antenna discrimination implies that the DTTB receiver lacks these advanced features. The receiver may have a single, less sophisticated antenna or lack the signal processing capabilities necessary to filter out IMT interference effectively. As a result, in areas where the IMT network and DTTB signals overlap or conflict, the DTTB reception quality may be significantly degraded.

The DTTB antenna discriminations are modeled through Monte Carlo simulations. To represent scenarios with full DTTB antenna discrimination, the DTTB receiver antenna is oriented in the opposite direction of the IMT network. Conversely, for scenarios with no DTTB antenna discrimination, the DTTB receiver antenna is pointed toward the IMT network. Figure 3 illustrates the model configurations of the Monte Carlo analysis performed by means of SEAMCAT software version 5.4.2 for Scenario A.

#### 4.1.2. Scenario B

IMT-BS interfering with a portable DTTB indoor receiver, as depicted in Figure 2B.

Similar to the methodology described in scenario A, the Monte Carlo simulation method assesses the interference level and derives the minimum separation distance for compatibility based on the protection criterion of I/N = −10 dB, in rural and urban areas.

#### 4.1.3. Scenario C

IMT-UEs interfering with fixed outdoor DTTB receiver, as shown in Figure 2C.

Monte Carlo simulation method assesses the interference level and derives the minimum separation distance for compatibility based on the protection criterion of I/N = −10 dB, in rural and urban areas. The case when UEs transmit the maximum possible power level is considered.

#### 4.1.4. Scenario D 

IMT-UE interfering with portable DTTB indoor receiver, as shown in Figure 2D.

In rural and urban areas, the Monte Carlo simulation method is employed to assess the interference level for compatibility based on the I/N = −10 dB protection criterion. The analysis considers the worst-case scenario, where the UE operates at maximum power and is located in the same room within the building as the DTTB receiver.

### 4.2. Interferences from the DTTB to the IMT Systems

#### 4.2.1. Scenario E

IMT uplink reception is interfered with by the DTTB transmitter, as shown in Figure 2E.

In both rural and urban environments, the analysis focuses on the variation of average bitrate loss at the IMT BS with distance, considering different classes of DTTB transmitters, namely HTHP transmitters, −6 dB reduced power HTHP transmitters, and MTMP transmitters. The objective is to ensure adequate protection by deriving the required coordination distance so that the average bitrate loss at the IMT BS receiver remains below 5% [45,46]. The protection criterion used for this assessment is I/N = −6 dB. In this scenario, a load factor of 50% is considered for the IMT network, and the propagation is assumed to occur across a land path for different percentages of time (1%, 5%, 10%, and 50%).

Additionally, the impact of variation in the height of the interfered IMT BS caused by the DTTB HTHP transmitter is analyzed. The height varies from 30 to 20 to 15 m, and its influence on the average bitrate loss at the IMT BS receiver is examined as a function of the separation distance. Similar to the previous case, a load factor of 50% is considered for the IMT network, and propagation is assumed to occur across a land path for 1% of the time.

#### 4.2.2. Scenario F

IMT downlink reception is interfered with by the DTTB, as shown in Figure 2F.

In both rural and urban environments, the analysis focuses on the variation of average bitrate loss at the IMT UEs with distance from different classes of DTTB transmitters, considering the protection criterion of I/N = −6 dB. The objective is to derive the required coordination distance to ensure adequate protection, ensuring that the average bitrate loss at the IMT UE receivers does not exceed 5%.

For this scenario, a load factor of 50% is considered for the IMT network, and the propagation is assumed to occur across a land path for different percentages of time (1%, 5%, 10%, and 50%). 

## 5. Results and Compatibility Analysis

This section presents the Monte Carlo simulation results of the co-channel interference between IMT and DTTB systems in the 470–694 MHz band following the methodology described earlier.

### 5.1. Interferences from the IMT Systems to the DTTB

#### 5.1.1. Scenario A

In order to protect a DTTB rooftop receiver from IMT BSs, the corresponding minimum separation distance under the protection criterion of I/N = −10 dB, is presented in detail.

Table 5 and Table 6 report the minimum coordination distance (in kilometers) between a network of seven tri-sectorized IMT BSs and the rooftop DTTB receiver with/without antenna discrimination for protection criterion of I/N= −10 dB, in rural and urban areas, respectively. Two different load factors are considered (50% and 20%) for the IMT network and four different times for the propagation model (1.75%, 5%, 10%, and 50%) across 100% land path. 

The results show the variation of the minimum separation distances required according to the parameters considered in the simulations. For instance, when considering an IMT network with a load factor of 50% and a fixed DTTB outdoor receiver with full antenna discrimination at 1.75% of the time, a separation distance of 83 km is necessary to meet the protection criterion of I/N = −10 dB. Limiting the interfering IMT BS coverage radius in urban environments to 1.5 km instead of 8 km in rural environments results in lower required separation distances under the same conditions.

To further illustrate the impact of separation distance, Figure 4 and Figure 5 provide a comparison of the probability of exceeding the I/N = −10 dB threshold for a given separation distance in rural and urban environments, respectively. These figures consider a scenario where a DTTB receiver is interfered with by seven cellular sites with a 50% network loading at 1.75% of the time, with and without antenna discrimination. 

The findings presented in Figure 4 and Figure 5 provide a clear understanding of the impact of separation distance on the probability of exceeding the specified interference threshold in different settings. At a 5% probability that I/N = −10 dB is exceeded, the results indicate the following minimum separation distances for a DTTB receiver with a 50% network loading and operating at 1.75% of the time: Rural DTTB receiver: A separation distance of 83 km is required with full antenna discrimination, while a separation distance of 290 km is needed without antenna discrimination.Urban DTTB receiver: A separation distance of 37 km is necessary with full antenna discrimination, whereas a separation distance of 167 km is required without antenna discrimination.

Furthermore, the study also evaluated the interference probability and the influence of various parameters and assumptions on the coordination distance between a seven-BS tri-sectorized IMT network and a DTTB rooftop receiver in a rural area. The analysis considered a DTTB receiver benefiting from full antenna discrimination and an IMT network with a load factor of 50%.

Table 7 presents the results regarding the interference probability in a DTTB rooftop receiver interfered with by varied TV Effective Isotropic Radiated Power (EIRP) and TV coverage areas representing different classes of DTTB transmitters.

In this scenario, there is no effect of using different classes of DTTB transmitters, since the same interference probability is obtained. This can be justified by the fact that TV receivers located at the edge of the TV coverage area maintain equivalent I/N ratios across various classes of DTTB transmitters.

Table 8 presents the minimum coordination distance (in kilometers) with a restriction on IMT-BS transmission power.

In rural areas, Figure 6 provides a comparison of interference probabilities between IMT-BS and a stationary outdoor DTTB receiver at varying separation distances with the limitation on IMT-BS transmission power. Full antenna discrimination, network loading of 50%, and a time allocation of 1.75% are considered.

At a 5% probability that I/N= −10 dB is exceeded, the minimum separation distance between the IMT network and the rural DTTB receiver is 50 km, given that the IMT-BS transmit power limit is 44.3 dBm. This contrasts with a separation distance of 83 km when the IMT-BS transmit power is 58 dBm.

Table 9 and Figure 7 illustrate the influence of the antenna tilt angle in the IMT BS on the probability of interferences in the fixed outdoor DTTB receiver.

It has been observed that as the antenna tilt in the IMT BS increases from 0 to −7 degrees, the interference probability for the same separation distance of 83 km from the outdoor DTTB receiver decreases. When adhering to recommended practices and aiming to protect 95% of locations with a probability of interference at 5%, the minimum separation distance required is 76 km for an antenna tilt of −7 degrees, in contrast to 83 km for an antenna tilt of −3 degrees. 

#### 5.1.2. Scenario B

To ensure the protection of a portable DTTB indoor receiver from IMT BSs, the corresponding minimum separation distance under the protection criterion of I/N= −10 dB, is shown in detail. 

Table 10 and Table 11 report the minimum coordination distance (km) between a network of seven IMT tri-sectorized BSs and an indoor DTTB portable receiver for the protection criterion of I/N = −10 dB, in rural and urban areas, respectively. Two different load factors are considered (50% and 20%) for the IMT network and four different times for the propagation model (1.75%, 5%, 10%, and 50%) across 100% land path. 

In this particular scenario, it is observed that shorter separation distances are necessary to fulfil the protection criteria. This can be attributed to the fact that the IMT downlink interfering signal experiences additional building penetration loss, which has a minimal impact on an indoor DTTB portable receiver. Hence, it is logical that the required separation distances are reduced in order to meet the specified protection criteria.

#### 5.1.3. Scenario C

To ensure the protection of a DTTB rooftop receiver from IMT-UEs, the corresponding minimum separation distance under the protection criterion of I/N= −10 dB, is shown in detail.

Table 12 presents the minimum coordination distance between IMT-UEs and a DTTB rooftop receiver, considering the antenna discrimination and the protection criterion of I/N = −10 dB. The scenario considers a load factor of 50%. 

Despite the DTTB rooftop receiver being positioned at the edge of the IMT network’s coverage, the interference from the IMT UEs to the DTTB receiver is negligible. This holds true in border areas, making it relatively effortless to achieve effective protection measures.

#### 5.1.4. Scenario D

Even when considering the worst case where an LTE-UE is located in the same room as the DTTB receiver and transmitting at maximum power, the interference remains minimal. In practice, user equipment seldom operates at maximum power. Additionally, it is worth noting that this coexistence scenario is not deemed critical in border areas.

### 5.2. Interferences from the IMT Systems to the DTTB

#### 5.2.1. Scenario E

The obtained results reveal the minimum separation distances required between DTTB transmitters and the IMT BS receiver operating in the 470–694 MHz band. These distances are necessary to ensure the desired performance of the IMT network while adhering to the protection criterion of I/N = −6 dB and maintaining an average bitrate loss of no more than 5% at the IMT BS receiver.

Figure 8 illustrates the variation of average bitrate loss at the IMT BS with distance from different classes of DTTB transmitters in rural environments. Similarly, Figure 9 depicts this variation in urban environments. Both figures consider a load factor of 50% at 1% of the time across a 100% land path.

As anticipated, the average bitrate loss experienced at the IMT BS receiver tends to increase as the transmit power of the DTTB signal becomes stronger. However, adhering to the prescribed separation distances between the DTTB transmission tower and the IMT BSs can mitigate this increase in bitrate loss, making it possible to achieve an average bitrate loss of less than 5% at the IMT BS receiver.

Table 13 and Table 14 provide detailed information on the coordination distances required to protect the IMT BS receiver from different classes of DTTB transmitters in rural and urban areas, respectively. These data adhere to the protection criterion of I/N = −6 dB and average bitrate loss below 5% at the IMT BS receiver.

The Monte Carlo simulation results demonstrate that the interference from the DTTB transmitter to the IMT-BS represents the most challenging scenario, requiring the longest coordination distances to maintain the desired performance level of the IMT-BS receiver. To achieve an average bitrate loss of less than 5% for 50% of the time across a land path in rural areas, the separation distance between the MTMP DTTB transmitter and the IMT BS needs to exceed 175 km. Similarly, in the case of the reduced power (−6 dB) HTHP DTTB transmitter, the separation distance should be more than 273 km, compared to 320 km for the full-power HTHP DTTB transmitter. It should be noted that for higher time availability across land paths, the separation distance needs to be increased accordingly.

The coordination distances required to protect the IMT BS receiver from the MTMP DTTB transmitter under the protection criterion of I/N = −6 dB and for an average bitrate loss below 5% at the IMT BS receiver, depending on BS height in a rural environment, are given in Table 15. 

The results show that the variation in the IMT BS antenna height causes a remarkable effect on the results in this scenario. It is observed that the minimum required separation distance between an IMT BS with a height of 20 m and an MTMP DTTB transmitter, while ensuring an average bitrate loss of 5% for 50% of the time across a land path, is approximately 156 km. This distance is lower than the coordination distance of 177 km for an IMT BS with a height of 30 m. Thus, adjusting the height of the IMT-BS to a value lower than 30 m can effectively reduce the minimum separation distances required.

#### 5.2.2. Scenario F

Figure 10 and Figure 11 provide valuable insights into the variation of average bitrate loss at the IMT UE receivers with distance from different classes of DTTB transmitters in rural and urban environments, respectively. These figures specifically consider the protection criterion of I/N = −6 dB and an average bitrate loss below 5%.

As expected, the average bitrate loss experienced at the IMT UE receiver tends to increase as the transmit power of the DTTB signal becomes stronger. However, by adhering to the prescribed separation distances between the DTTB tower and the IMT UEs, an average bit rate loss of less than 5% can be achieved at the IMT UE receivers.

Table 16 and Table 17 report in-depth information on the coordination distances necessary to protect the IMT UE receivers from various classes of DTTB transmitters in rural and urban environments, respectively. 

These data adhere to a protection criterion of I/N = −6 dB, and its primary goal is to ensure that the average bitrate loss experienced at the IMT BS receiver remains below the 5% threshold.

The findings reveal that shorter separation distances are required to protect IMT UE receivers from DTTB transmitter transmissions. As the DTTB transmit power increases, the minimum required separation distances also increase to meet the specified protection criteria.

Moreover, the results demonstrate that the required separation distances are generally shorter in urban environments compared to rural environments when the same conditions are applied. This observation can be attributed to the fact that in urban environments a higher percentage (70%) of IMT UE receivers are expected to be indoors, while in rural environments the distribution is more balanced, with 50% indoor and 50% outdoor IMT UE receivers. As a result, the interfering DTTB transmitter signal in urban areas experiences additional building penetration loss, which has a minimal impact on IMT UE indoor receivers. These variations in environmental characteristics contribute to the differences in the required separation distances between the two settings.

## 6. Conclusions

This contribution presents co-channel sharing and compatibility studies between the IMT-2020 system under the mobile service and the DTTB system under the broadcasting service in the 470–694 MHz frequency band. The studies were performed using the Monte Carlo simulations method by means of the SEAMCAT software.

The separation distances required between both services were derived based on protection criteria in rural and urban environments. Additionally, in order to consider the most interesting cases from the point of view of administrations, the influence of several parameters on the required separation distances was examined.

Regarding the interferences from the IMT systems to the DTTB, it is concluded that:For fixed outdoor DTTB reception interfered with by IMT-BSs, it is observed that:

With the recommended practice parameters, which aim to protect 95% of locations, it was found that the separation distance between an IMT BS network and a fixed outdoor DTTB receiver located at the edge of DTTB coverage is typically in the range of tens of kilometers for a medium load factor (50%) and slightly less for a low load factor (20%). These distances assume that the DTTB receiver’s antenna benefits from full discrimination. By implementing a small buffer zone between the two systems, these coordination distances can be achieved.

However, when no antenna discrimination is present, significantly larger separation distances exceeding 100 km are required. In such cases, achieving coordination distances becomes more challenging, thus disrupting DTTB service in the affected area.

The use of different classes of DTTB transmitters does not have a notable effect on the interference probability from IMT-BSs to fixed outdoor DTTB reception. However, reducing the maximum EIRP of IMT-BSs and adjusting the antenna tilt can significantly reduce interferences and subsequently decrease the required separation distances for fixed outdoor DTTB reception.
For portable indoor DTTB reception interfered with by IMT-BSs, it is observed that:

Thanks to the wall penetration loss, the effect of interference from IMT-BS on indoor portable reception is minimal. As a result, shorter separation distances between the IMT-BSs and the portable indoor DTTB receivers are required compared to fixed outdoor DTTB reception. This enables coexistence between the IMT and DTTB systems, where indoor DTTB reception is more common, particularly between neighboring countries.
For fixed outdoor DTTB reception interfered with by IMT-UEs, it is observed that:

The interference from the IMT-UEs to the rooftop DTTB receiver is much less than that from the BSs. Thus, achieving this coexistence scenario in the border areas is relatively easier.
For portable indoor DTTB reception interfered with by IMT-UEs, it is observed that:

The interference from the IMT-UE to the portable indoor DTTB receiver was also very small, even in the worst cases. However, this coexistence scenario is not considered decisive in border areas.

Regarding the interferences from the DTTB to the IMT systems, it is concluded that:For IMT uplink reception interfered with by a DTTB transmitter, it is observed that:

Adhering to the recommended practice guidelines to maintain optimal performance of the IMT-BS receiver, which entails achieving an average bit rate loss of less than 5%, necessitates maintaining significant separation distances between the DTTB transmitter and the IMT-BS receiver, spanning hundreds of kilometers. In this case, the interferences become challenging to avoid, impeding the implementation of IMT services in the affected region.

Unsurprisingly, the average bitrate loss at the IMT BS receiver increases with the interfering transmitter’s increasing power and/or height. The results show that interference from a full-power HTHP DTTB transmitter may restrain the IMT channel capacity. Conversely, reducing the height of the IMT-BS antenna mitigates the impact of interference from the DTTB transmitter.

Therefore, reducing the transmission power of the DTTB system and lowering the height of the IMT-BS antenna emerge as critical factors in minimizing interference and limiting the distance to protect the IMT uplink reception interfered by DTTB transmitter.
For IMT downlink reception interfered with by a DTTB transmitter, it is observed that:

To ensure the protection of IMT-UE receivers from the influence of DTTB transmitters, it is necessary to maintain separation distances ranging in the tens of kilometers. These minimum separation distances increase in accordance with the increased transmit power and/or height of the DTTB system to meet the protection criteria. 

In urban areas, where approximately 70% of IMT-UE receivers are expected to be indoors, the DTTB transmitter’s interfering signal encounters additional loss due to building penetration. As a result, the required separation distances in urban settings tend to be shorter, considering the specific conditions and the need to address potential interferences.

Ultimately, the simulation results confirm the conclusions drawn from earlier studies conducted on previous digital dividend bands that sharing the same band between IMT and DTTB is very difficult and will result in significant mutual interference, making this operation impractical. 

Planned future studies will explore the possibility of the coexistence of IMT and DTTB systems in the adjacent channels in the 470–694 MHz band.

## Figures and Tables

**Figure 1 sensors-23-08714-f001:**
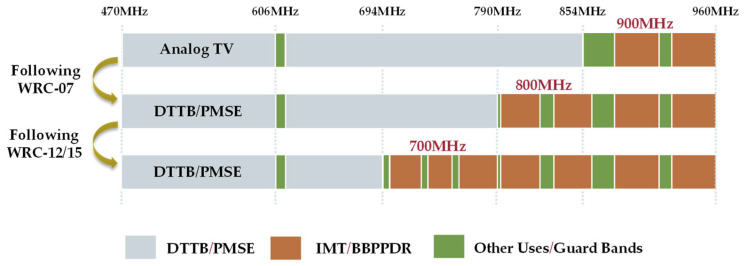
Radiofrequency spectrum allocation between over-the-air services following WRCs.

**Figure 2 sensors-23-08714-f002:**
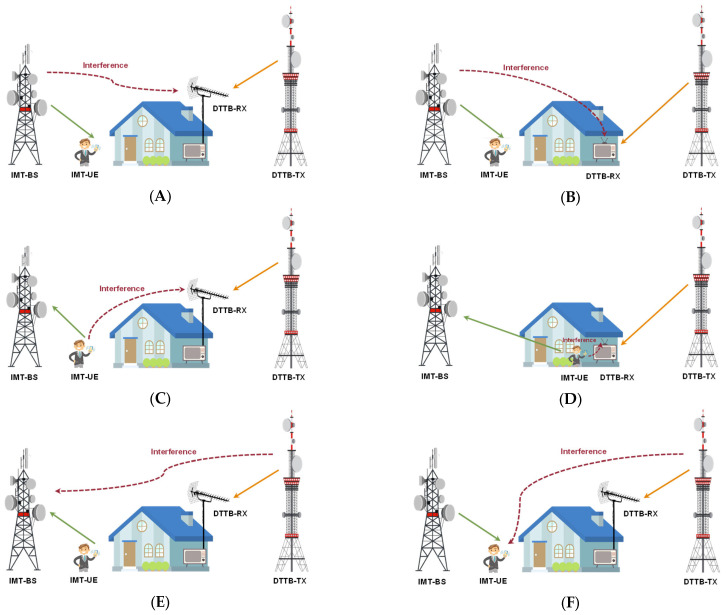
Potential Coexistence Scenarios between IMT and DTTB in the 470–694 MHz Band. (**A**) IMT-BS Interfering with Fixed Rooftop DTTB Reception; (**B**) IMT-BS Interfering with Portable DTTB Indoor Reception; (**C**) IMT-UE Interfering with Fixed Rooftop DTTB Reception; (**D**) IMT-UE Interfering with Portable DTTB Indoor Reception; (**E**) DTTB Transmitter Interfering with IMT-BS; (**F**) DTTB Transmitter Interfering with IMT-UE.

**Figure 3 sensors-23-08714-f003:**
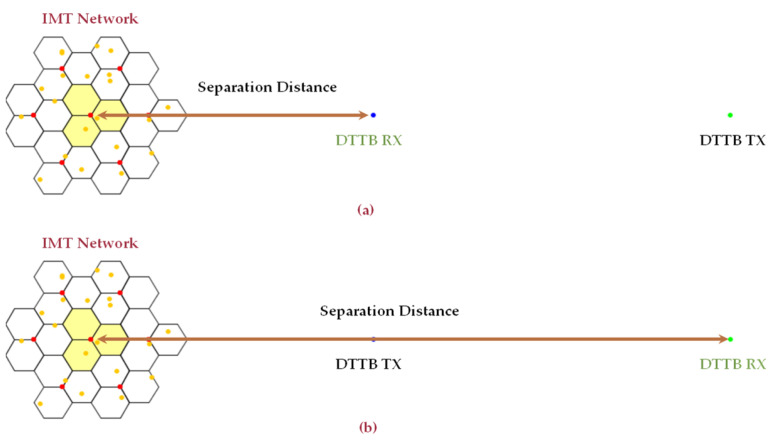
Model configuration for IMT network interfering with DTTB receiver. (**a**) The case of full DTTB antenna discrimination. (**b**) The case of no DTTB antenna discrimination.

**Figure 4 sensors-23-08714-f004:**
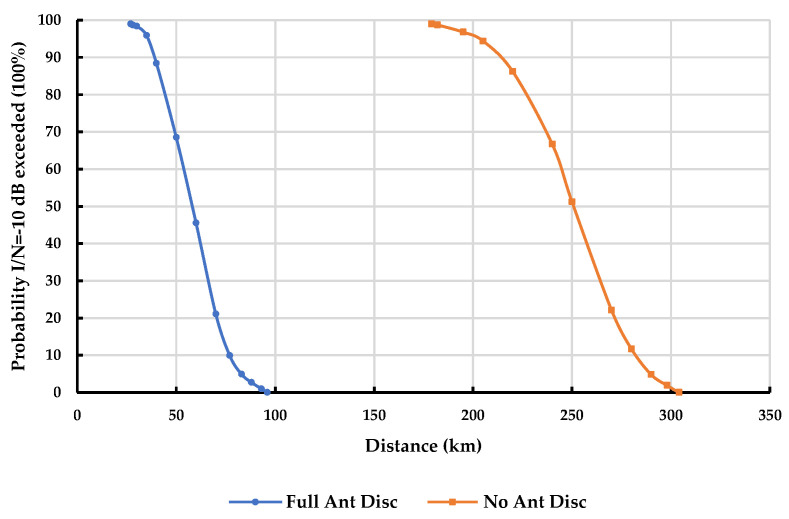
Probability I/N= −10 dB exceeded for a given distance at 1.75% time in the DTTB rural case with/without antenna discrimination—Network loading 50%.

**Figure 5 sensors-23-08714-f005:**
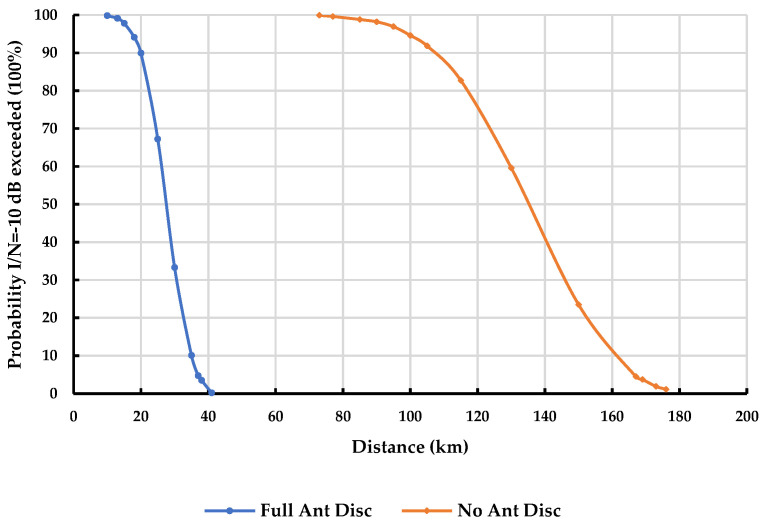
Probability I/N= −10 dB exceeded for a given distance at 1.75% time in the DTTB urban case with/without antenna discrimination—Network loading 50%.

**Figure 6 sensors-23-08714-f006:**
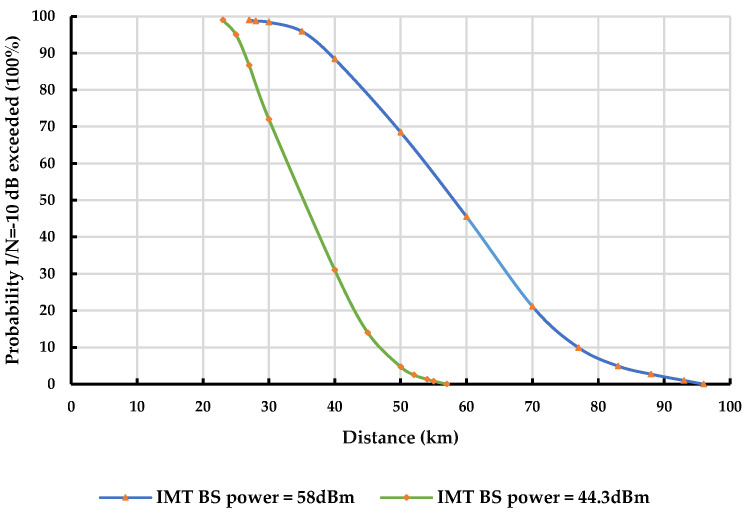
IMT-BS interference probabilities with a fixed outdoor DTTB receiver according to separation distances in rural areas, simulations with limitation of IMT-BS transmission power.

**Figure 7 sensors-23-08714-f007:**
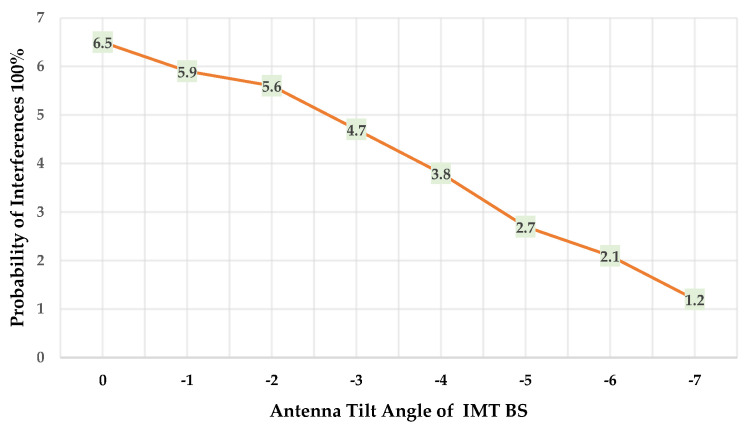
Probability of interferences as a function of the antenna tilt.

**Figure 8 sensors-23-08714-f008:**
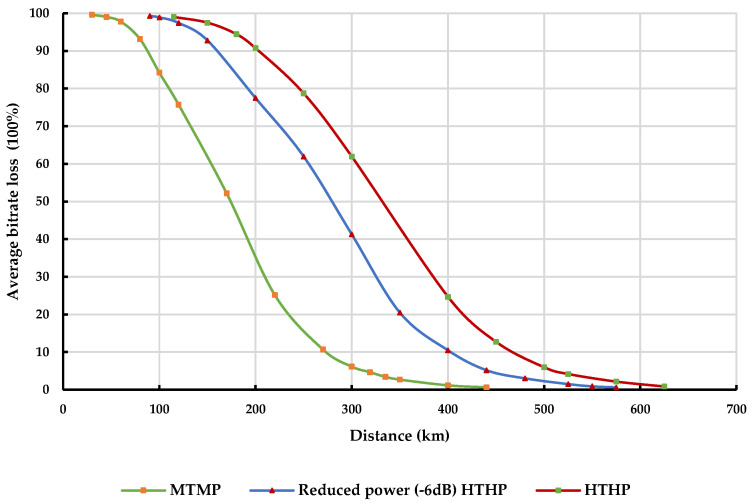
Average bitrate loss at IMT BS Rx due to different classes of DTTB transmitters in a rural environment.

**Figure 9 sensors-23-08714-f009:**
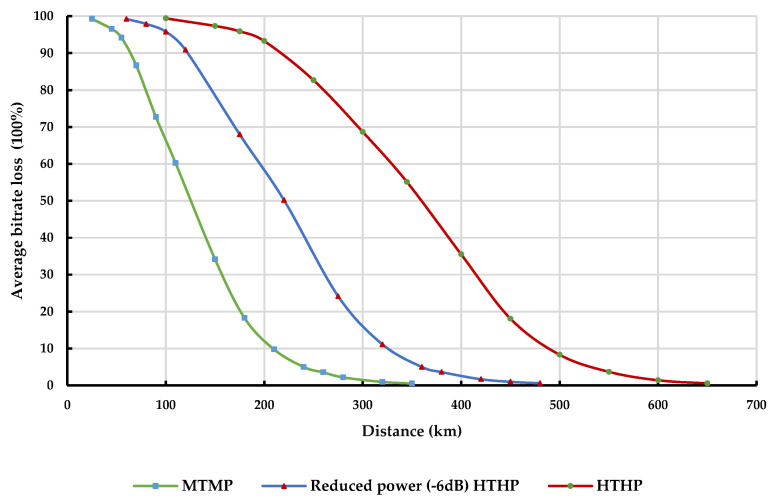
Average bitrate loss at IMT BS Rx due to different classes of DTTB transmitters in an urban environment.

**Figure 10 sensors-23-08714-f010:**
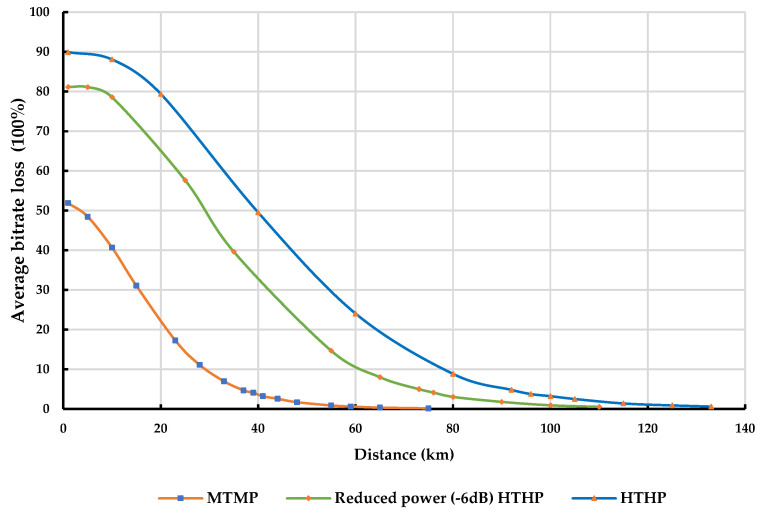
Average bitrate loss at IMT UE Rx due to different classes of DTTB transmitters in a rural environment.

**Figure 11 sensors-23-08714-f011:**
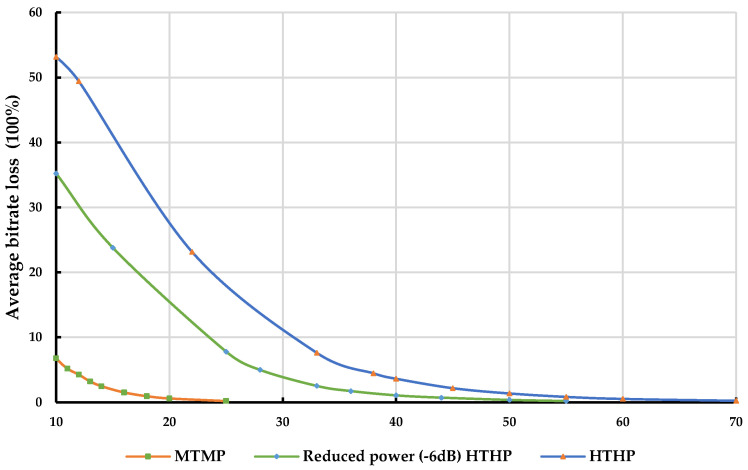
Average bitrate loss at IMT UE Rx due to different classes of DTTB transmitters in an urban environment.

**Table 1 sensors-23-08714-t001:** Approaches to study the coexistence of DTTB and IMT systems [9].

Approach	Purpose	Advantages	Limitations
Link budget analysis	Calculate the power budget for DTTB and IMT systems.	- Provides theoretical foundation. - Quick and cost-effective.	- May not account for real-world interference. - Assumes ideal conditions.
Simulations	Predict how DTTB and IMT systems interact in different scenarios.	- Allows detailed scenario analysis. - “What-if” analysis. - Cost-effective.	- Accuracy depends on model quality. - Real-world may differ from assumptions.
Laboratory measurements	Controlled testing under controlled conditions.	- Precise control over variables. - Validates theoretical models.	- May not fully replicate real-world conditions. - May not cover all scenarios.
Field measurements	Collect real-world data.	- Provides real-world insights. - Identifies real-world issues.	- Time-consuming and expensive. - Limited control over variables. - May not cover all scenarios.

**Table 2 sensors-23-08714-t002:** IMT-BS parameters applied in the simulations for bands below 1 GHz [44].

Base Station Characteristics Cell Structure	Urban Macro	Rural Macro
Cell radius	0.5–5 km (typical value to be used in sharing studies for urban macro 1.5 km)	>5 km (typical value to be used in sharing studies 8 km)
Antenna height	30 m, 20 m	
Sectorization	3 sectors	
Antenna downtilt	Range from 0 to −7 degrees (typical value to be used in sharing studies −3 degrees)	
Frequency reuse	1	
Configuration of interfering sources	7 tri-sectorized base stations	
Antenna pattern	Rec. ITU-R F.1336 (Rec. 3.1) ka = 0.7 kp = 0.7 kh = 0.7 kv = 0.3 Horizontal 3 dB beam width: 65° Vertical 3 dB beam width: determined from horizontal beam width (Rec. ITU-R F.1336) or actual antenna data.	
Tx Antenna orientation	Sector pointing direction based on 3GPP Tri-sector deployment	
Antenna polarization	Linear/±45°	
Feeder loss	3 dB	
Typical channel bandwidth	10 MHz	
Maximum BS output power (Report ITU-R M.2292)	46 dBm in 10 MHz	
Maximum BS antenna gain (Report ITU-R M.2292)	15 dBi	
Maximum BS output power/sector (EIRP)	58 dBm baseline value/44.3 dBm results from Gaussian distribution used to simulate the IMT-BS power variation in time.	
Network loading factor	20%, 50%	
TDD/FDD/SDL	FDD/SDL	

**Table 3 sensors-23-08714-t003:** IMT-UE parameters applied in the simulations for bands below 1 GHz [44].

User Terminal Characteristics	Urban Macro	Rural Macro
Indoor UE usage (Report ITU-R M.2292)	70%	50%
Indoor UE penetration loss	Recommendation ITU-R P.2109	
UE density for simultaneous transmission	3 UEs/sector	
UE height	1.5 m	
Avg UE output power	Transmit power control (TPC) utilization	
Typical UE antenna gain	−3 dBi	
Body loss	4 dB	
Power control model	Rec. ITU-R M.2101	
Maximum UE output power	23 dBm	
Power target per RB	−92.2 dBm	
Path loss compensation factor	0.8 dB	

**Table 4 sensors-23-08714-t004:** DTTB system parameters applied in the simulations for the 470–694 MHz band [45,46].

DTTB Characteristics				
Centre frequency	600 MHz			
Channel BW	8 MHz			
Feeder loss	4 dB			
Noise figure	6 dB			
Cell edge coverage probability	95%			
**DTTB Transmitter Characteristics**				
Classes of DTTB Tx	**HTHP**	**Reduced Power (−6dB) HTHP**		**MTMP**
ERP/e.i.r.p.	83/85.15 dBm	77/79.15 dBm		67/69.15 dBm
Coverage radius	74.8 km	74.8 km		38 km
Effective antenna height	300 m	300 m		150 m
Antenna height above ground level (a.g.l.)	200 m	200 m		75 m
Tx antenna	10 dBi	10 dBi		10 dBi
Antenna pattern—horizontal	Omnidirectional	Omnidirectional		Omnidirectional
Antenna pattern—vertical antenna aperture	Using a 24λ aperture with 1° beam tilt	Using a 24λ aperture with 1° beam tilt		Using a 16λ aperture with a 1.6° beam tilt
**DTTB Receiver Characteristics**				
DTTB reception modes	**Fixed Outdoor Reception**			**Portable Indoor Reception**
Receiver antenna height	10 m			1.5 m
Receiver antenna gain	−6.85 dBi (Full Discrimination) Report ITU-R BT.2383 (Results of 9.15–16 dBi)			2.15 dBi
	9.15 dBi (No Discrimination)			
Building entry loss	-			10 dB (Rural)
				18.14 dB (Urban)
X-Pol. discrimination	−3 dB (Only applicable in no discrimination case)			
Rx antenna pattern	ITU-R BT.419.13			
DTTB receiver location	At the DTTB coverage edge			
Placement	100 m × 100 m at cell edge			

**Table 5 sensors-23-08714-t005:** Minimum coordination distance between a seven-BS tri-sectorized IMT network and DTTB rooftop receiver with/without antenna discrimination for I/N = −10 dB in a rural area under different load factors and percentage of the time.

Antenna Discrimination	Load Factor	Minimum Coordination Distance (km) at X% of Time			
		1.75%	5%	10%	50%
Full Ant. Disc.	50%	83	75	70	63
	20%	73	66	62	58
No Ant. Disc.	50%	290	255	231	173
	20%	266	234	212	157

**Table 6 sensors-23-08714-t006:** Minimum coordination distance between a seven-BS tri-sectorized IMT network and DTTB rooftop receiver with/without antenna discrimination for I/N = −10 dB in an urban area under different load factors and percentage of the time.

Antenna Discrimination	Load Factor	Minimum Coordination Distance (km) at X% of Time			
		1.75%	5%	10%	50%
Full Ant. Disc.	50%	37	34	31	29
	20%	33	30	28	26
No Ant. Disc.	50%	167	145	131	93
	20%	146	126	115	84

**Table 7 sensors-23-08714-t007:** IMT interference with DTTB rooftop receiver: simulations with variable EIRP and TV coverage area.

Rural area Full Ant. Dis. LF 50%	IMT BS power = 58 dBm	**Classes of DTTB Transmitters**	**Time 1.75%**		
			**IP%**	**Active TV Users (%)**	**Separation Distance (km)**
		MTMP	4.9%	95.1%	83
		Reduced power (−6 dB) HTHP	4.7%	95.3%	83
		HTHP	4.8%	95.2%	83

**Table 8 sensors-23-08714-t008:** IMT interference with DTTB rooftop receiver: simulations with limitation of IMT-BS transmission power.

Rural area Full Ant. Dis. LF 50%	MTMP DTTB	**IMT BS Power**	**Minimum Coordination Distance (km) at X% of Time**	
			**1.75%**	**50%**
		58 dBm	83	63
		44.3 dBm	50	45

**Table 9 sensors-23-08714-t009:** IMT interference with DTTB rooftop receiver: simulations with increasing IMT BS antenna tilt angle.

Rural area Full Ant. Dis. LF 50% Time 1.75%	MTMP DTTB	**Antenna Tilt Angle in IMT BS**	**IP% for Separation Distance = 83 km**
		0	6.5%
		−1	5.9%
		−2	5.6%
		−3	4.7%
		−4	3.8%
		−5	2.7%
		−6	2.1%
		−7	1.2%

**Table 10 sensors-23-08714-t010:** Minimum coordination distance between a seven-BS tri-sectorized IMT network and indoor DTTB portable receiver for I/N = −10 dB in a rural area under different load factors and percentage of the time.

Load Factor	Minimum Coordination Distance (km) at X% of Time			
	1.75%	5%	10%	50%
50%	48	46	45	43
20%	43	42	40	39

**Table 11 sensors-23-08714-t011:** Minimum coordination distance between a seven-BS tri-sectorized IMT network and indoor DTTB portable receiver for I/N = −10 dB in An Urban area under different load factors and percentage of the time.

Load Factor	Minimum Coordination Distance (km) at X% of Time			
	1.75%	5%	10%	50%
50%	19	18	17	16
20%	17	16	15	14

**Table 12 sensors-23-08714-t012:** Minimum coordination distance between IMT UEs and DTTB rooftop receiver with/without antenna discrimination for I/N = −10 dB in a rural area under 50% load factor.

Antenna Discrimination	Load Factor	Minimum Coordination Distance (km) at X% of Time	
		1%	50%
Full Ant. Disc.	50%	Coverage edge	Coverage edge
No Ant. Disc.	50%	Coverage edge	Coverage edge

**Table 13 sensors-23-08714-t013:** Minimum distances between different classes of DTTB transmitters and IMT BS receivers for an average bitrate loss <5% in a rural environment.

Classes of DTTB Transmitters	Minimum Coordination Distance (km) at X% of Time			
	1%	5%	10%	50%
MTMP	319	255	230	177
Reduced power HTHP	>400	375	339	273
HTHP	>500	437	405	320

**Table 14 sensors-23-08714-t014:** Minimum distances between different classes of DTTB transmitters and IMT BS receivers for an average bitrate loss <5% in an urban environment.

Classes of DTTB Transmitters	Minimum Coordination Distance (km) at X% of Time			
	1%	5%	10%	50%
MTMP	240	197	179	135
Reduced power HTHP	360	303	273	213
HTHP	>450	359	325	256

**Table 15 sensors-23-08714-t015:** Minimum distances between the MTMP DTTB transmitter and IMT BS receivers for an average bitrate loss <5% depending on BS height in a rural environment.

MTMP DTTB	**Base Station Height**	**Minimum Coordination Distance (km) at X% of Time**	
		**1%**	**50%**
	30 m	319	177
	20 m	284	156
	15 m	255	139

**Table 16 sensors-23-08714-t016:** Minimum separation distances between different classes of DTTB transmitters and IMT UE receivers for an average bitrate loss <5% in a rural environment.

Classes of DTTB Transmitters	Minimum Coordination Distance (km) at X% of Time			
	1%	5%	10%	50%
MTMP	37	36	36	36
Reduced power HTHP	73	67	64	63
HTHP	92	81	77	74

**Table 17 sensors-23-08714-t017:** Minimum separation distances between different classes of DTTB transmitters and IMT UE receivers for an average bitrate loss <5% in an urban environment.

Classes of DTTB Transmitters	Minimum Coordination Distance (km) at X% of Time			
	1%	5%	10%	50%
MTMP	12	11	11	11
Reduced power HTHP	28	28	27	27
HTHP	38	36	35	35

## Data Availability

Data are available upon request.

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
