# Peer review of "Co-Channel Compatibility Analysis of IMT Networks and Digital Terrestrial Television Broadcasting in the Frequency Range 470–694 MHz Based on Monte Carlo Simulation"

_sensors, 2023, doi:10.3390/s23218714_

Round 1

Reviewer 1 Report

1.     In this paper, a novel technique, named index modulation, is very useful for the co-channel compatibility, where only partial channels are used for transmission. Hence, to largely improve the quality of this paper, it is supposed to provide the index modulation [1-2] in Introduction.

[1] Index Modulation Multiple Access for 6G Communications: Principles, Applications, and Challenges, IEEE Network, 2023.

[2] A Survey on Spatial Modulation in Emerging Wireless Systems: Research Progresses and Applications, IEEE JSAC, 2019.

2.     Please clarify the difference of two models in Fig. 3,

3.     It seems like the configurations of Figs. 4 and 5 are the same.

4.     The complexity of co-channel sharing and compatibility should be measured in this paper.

5.     Please draw all curves smoother. In addition, please improve the resolution of all figures.

6.     Please correct some typo errors.

 Please correct some typo errors.

Author Response

Response to the Reviewer 1

We thank Reviewer 1 for evaluating our article and insightful suggestions that helped us make the paper much better. Our answers to Reviewer 1 remarks are listed below.

Note:  The major modifications are marked in green color in the revised manuscript.

Point 1. In this paper, a novel technique, named index modulation, is very useful for co-channel compatibility, where only partial channels are used for transmission. Hence, to largely improve the quality of this paper, it is supposed to provide the index modulation [1-2] in the Introduction.

[1] Index Modulation Multiple Access for 6G Communications: Principles, Applications, and Challenges, IEEE Network, 2023.

[2] A Survey on Spatial Modulation in Emerging Wireless Systems: Research Progresses and Applications, IEEE JSAC, 2019.

Response: Thank you for drawing our attention to emphasize the significance of incorporating index modulation technology alongside other strategies to enhance co-channel compatibility.

In the revised manuscript (specifically in the introduction from line 169 to line 182), we added a detailed paragraph about index modulation. This addition emphasizes the significance of incorporating index modulation technology alongside other strategies to enhance co-channel compatibility within wireless communication systems. We have demonstrated how index modification can be utilized to improve coexistence in various ways. Furthermore, we have highlighted the complexities associated with integrating index modulation techniques into the co-channel operation between IMT and DTTB systems operating within the 470-694 MHz frequency range. This integration necessitates careful consideration of design, coordination, and adaptation mechanisms. The specific approach chosen will depend on system requirements, the nature of interference, and the available resources.

Point 2. Please clarify the difference of two models in Fig. 3

Response: Thank you for your drawing our attention to the fact, that the difference of the two models were not clear in the original text and suggesting clarifying the difference between the model configurations (full DTTB antenna discrimination and no DTTB antenna discrimination).

In the revised manuscript (from line 362 to line 380), we have clarified that the primary difference between these model configurations lies in the receiver's ability to discriminate between the DTTB signal and IMT network interference. We underscored that the selection of configuration has a significant impact on the viewer's experience in areas where these signals overlap. Furthermore, we provided how these configurations are modelled through Monte Carlo simulations to represent scenarios with full DTTB antenna discrimination and scenarios without DTTB antenna discrimination.

Point 3. It seems like the configurations of Figs. 4 and 5 are the same.

Response: Thank you for your comment.

The goal of both Figures 4 and 5 is to compare the probability of exceeding the I/N = –10 dB threshold for a given separation distance. Both figures address a scenario where a DTTB receiver is interfered with by 7 cellular sites with a 50% network loading at 1.75% of the time, with and without antenna discrimination. The results presented in Figures 4 and 5 provide a clear insight into how separation distance influences the likelihood of exceeding the specified interference threshold in different contexts.

The difference between these two figures is that Figure 4 illustrates the results in a rural environment, while Figure 5 illustrates the results in an urban environment. As detailed in the revised manuscript (from line 472 to line 477), we highlighted that rural environments require greater separation distances compared to urban environments.

Point 4. The complexity of co-channel sharing, and compatibility should be measured in this paper.

Response: Thank you for drawing our attention to this point.

A validation comparison analysis based on real measurements with real devices would undoubtedly strengthen the work.

However, such measurements are difficult to conduct in the field. We have consulted about this matter with the National Media and Infocommunications Authority in Hungary, but until now, specialists have not been able to organize such measurements in the field and even in the laboratory due to the large power and distances that cannot be simulated.

Also, setting up and maintaining a real-world test environment with actual devices can be prohibitively expensive and time-consuming due to the need for specialized equipment, licensing, and manpower. On the other hand, the limited statistical foundation of field measurement campaigns prevents drawing significant conclusions.

In this contribution, we have chosen to use simulations, in particular through the specialized software tool SEAMCAT, to investigate coexistence issues between DTTB and IMT systems for several compelling reasons: It offers a cost-effective, safe, controlled, repeatable, and flexible way to assess system interactions, accelerating the analysis process and providing valuable insights for decision-making and system design. Simulations grant complete control over test scenarios, allowing the creation of a wide range, including worst-case scenarios, which may be difficult or impossible to conduct in the real world. This control facilitates comprehensive testing and analysis. Additionally, international spectrum regulatory authorities follow the practice of simulating coexistence scenarios between DTTB and IMT systems, as specified in international agreements.

(We explained why we chose simulation in the revised manuscript from line 186 to line 194).

Point 5. Please draw all curves smoother. In addition, please improve the resolution of all figures.

Response: Thank you for your suggestion.

In the revised manuscript, we have redrawn all the curves to ensure they appear smoother and clearer. Furthermore, we have enhanced the resolution of the figures to the maximum achievable level. As for the resolution of curves, they were originally exported from Excel to Word and are, and as such, they are already in the main file format with the highest possible resolution.

Point 6. Please correct some typo errors.

Response: Thank you for your comment.

In the revised manuscript, we have corrected all typos and grammatical errors.

===========================

==================

=======

Reviewer 2 Report

This study specifically focuses on evaluating electromagnetic compatibility in potential co-channel sharing scenarios between Digital Terrestrial Television Broadcasting (DTTB) and International Mobile Telecommunications (IMT) systems within the 470-694 MHz frequency band that may arise in the foreseeable future. To assess the conditions for coexistence, a Monte Carlo 18 simulation method implemented through the SEAMCAT software is employed, examining six potential interference scenarios. The present study investigates the impact of various factors such as transmitter power, antenna heights, coverage radius, antenna discrimination, and antenna tilt angle on the separation distances. The focus lies primarily on critical interference scenarios across neighboring countries' borders. The simulation outcomes confirm that sharing the same frequency band between IMT and DTTB networks would result in significant mutual interference. Nevertheless, carefully analyzing diverse parameters and assumptions helped provide recommendations to reduce the required separation distances. Authors, claim that these findings are valuable for broadcasters, mobile operators, and regulators in establishing the technical coexistence conditions for DTTB and IMT in the new band. 

In my opinion this work lacks of novelty:  the topic has been extensively studied and coexistence in the 700MHz band is one of the more debated issue between broadcast and broadband mobile communities.  Moreover, simulations and propagation models used are reliable but limited and a validation comparison analysis based on real measurements with real devices (extensively available in literature) would give more strength to the work.

I suggest to the authors to review the literature, identifying published works that propose field tests and to make a comparison analysis to validate the results presented in the current form of the manuscript.

Author Response

Response to the Reviewer 2

We thank Reviewer 2 for evaluating our article and giving insightful suggestions that helped us improve the manuscript. Our answers to Reviewer 2 remarks are listed below.

Note:  The major modifications are marked in green color in the revised manuscript.

Point 1.  This study specifically focuses on evaluating electromagnetic compatibility in potential co-channel sharing scenarios between Digital Terrestrial Television Broadcasting (DTTB) and International Mobile Telecommunications (IMT) systems within the 470-694 MHz frequency band that may arise in the foreseeable future. To assess the conditions for coexistence, a Monte Carlo simulation method implemented through the SEAMCAT software is employed, examining six potential interference scenarios. The present study investigates the impact of various factors such as transmitter power, antenna heights, coverage radius, antenna discrimination, and antenna tilt angle on the separation distances. The focus lies primarily on critical interference scenarios across neighboring countries' borders. The simulation outcomes confirm that sharing the same frequency band between IMT and DTTB networks would result in significant mutual interference. Nevertheless, carefully analyzing diverse parameters and assumptions helped provide recommendations to reduce the required separation distances. Authors, claim that these findings are valuable for broadcasters, mobile operators, and regulators in establishing the technical coexistence conditions for DTTB and IMT in the new band. 

In my opinion this work lacks of novelty: the topic has been extensively studied and coexistence in the 700MHz band is one of the more debated issue between broadcast and broadband mobile communities.  Moreover, simulations and propagation models used are reliable but limited and a validation comparison analysis based on real measurements with real devices (extensively available in literature) would give more strength to the work.

Response: We would like to express our gratitude for evaluating our article and thank you for your suggestion.

Our manuscript presents a critical examination of electromagnetic compatibility issues arising from the co-channel sharing of DTTB and IMT systems within the new frequency range of 470-694 MHz. While the challenges of coexistence in the previous digital bands (700 MHz and 800 MHz) have been extensively discussed in the literature, our research holds unique significance and novelty.

Our contribution responds to the call from the International Telecommunication Union - Radiocommunication Sector (ITU-R) to conduct comprehensive sharing and compatibility studies, as mandated by Resolution 235 of WRC-15 and the preliminary agenda item 1.5 of the upcoming WRC-2023. What sets our work apart is that, as of now, there is only one published research addressing the evaluation of technical feasibility and requirements for harmonious coexistence in the new frequency band, 470-694 MHz. (This research is cited and referenced as reference [10] in our revised manuscript).

Our contribution is comprehensive, encompassing all potential interference scenarios contingent upon the type of IMT interfering link (Uplink or Downlink) and the DTTB reception type (fixed outdoor or portable indoor) in both rural and urban environments. This comprehensive approach has yielded valuable insights into interference dynamics. Furthermore, we meticulously analyzed various parameters and appropriate mitigation strategies, offering recommendations to reduce the required separation distances. The outcomes of our research hold great importance for spectrum regulators, broadcasters, mobile operators, and policymakers who are grappling with the complexities of spectrum allocation and coexistence between diverse services. The detailed insights and recommendations presented in our manuscript will contribute to making informed decisions and developing effective regulatory frameworks.

In contrast, most previous reference studies in previous digital bands typically considered specific scenarios and use cases. Also, they investigated only the effects of specific parameters.

Point 2.  I suggest to the authors to review the literature, identifying published works that propose field tests and to make a comparison analysis to validate the results presented in the current form of the manuscript.

Response: Thank you for this suggestion.

In the revised manuscript (specifically in the introduction from line 77 to line 182), we extensively reviewed the relevant literature according to the approaches used to investigate coexistence issues, which include link budget analysis, simulations, laboratory measurements, and field tests. We also identified published works that propose field tests to validate simulation results. We have included Table 1, which compares the purpose, advantages, and disadvantages of coexistence investigation approaches.

A validation comparison analysis based on real measurements with real devices would undoubtedly strengthen the work.

However, such measurements are difficult to conduct in the field. We have consulted about this matter with the National Media and Infocommunications Authority in Hungary, but until now, specialists have not been able to organize such measurements in the field and even in the laboratory due to the large power and distances that cannot be simulated.

Also, setting up and maintaining a real-world test environment with actual devices can be prohibitively expensive and time-consuming due to the need for specialized equipment, licensing, and manpower. On the other hand, the limited statistical foundation of field measurement campaigns prevents drawing significant conclusions.

In this contribution, we have chosen to use simulations, in particular through the specialized software tool SEAMCAT, to investigate coexistence issues between DTTB and IMT systems for several compelling reasons: It offers a cost-effective, safe, controlled, repeatable, and flexible way to assess system interactions, accelerating the analysis process and providing valuable insights for decision-making and system design. Simulations grant complete control over test scenarios, allowing the creation of a wide range, including worst-case scenarios, which may be difficult or impossible to conduct in the real world. This control facilitates comprehensive testing and analysis. Additionally, international spectrum regulatory authorities follow the practice of simulating coexistence scenarios between DTTB and IMT systems, as specified in international agreements.

(We explained why we chose simulation in the revised manuscript from line 186 to line 194).

=========================

=================

========

Reviewer 3 Report

An interesting topic is selected by the authors that is being persuaded by many researchers. However, the overall presentation of the work is not sufficient.

The novelty of the work is missing, authors described that in the abstract “Nevertheless, carefully analyzing diverse parameters and assumptions helped provide recommendations to reduce the required separation distances. These findings are valuable for broadcasters, mobile operators, and regulators in establishing the technical coexistence conditions for DTTB and IMT in the new band.” Is there a proof that shows these parameters were not being considered by the expert in the fields? It would be a good author to present a table of comparison of performance with currently used systems in literature and field and compare their performance with the proposed approach in the work and clearly show the difference of performance. The table of comparison can present the advantages and disadvantages of the proposed design approach compared to the currently used approaches.

The literature overview of the work must be improved. There are a total of 24 cited works, which is a very low citation number for such a hot topic. And moreover, there are no cited works from 2023 and only 2 cited works from 2022, and only 3 cited works from 2021.  For such a topic that is being actively persuaded by many researchers, this is a weakness for the work. The ratio of the recently cited works (after 2021) must be at least 25% of the total citation and the total citation for both literature overview and table of comparison must be around 40 or so. Authors are welcome to use the following related works on 3D printable FSS to their literature overview and table of comparison.

Author Response

Response to the Reviewer 3

We thank Reviewer 3 for evaluating our article and giving insightful suggestions that helped us improve the manuscript. Our answers to Reviewer 3 remarks are listed below.

Note:  The major modifications are marked in green color in the revised manuscript.

Point 1. An interesting topic is selected by the authors that is being persuaded by many researchers. However, the overall presentation of the work is not sufficient.

The novelty of the work is missing, authors described that in the abstract “Nevertheless, carefully analyzing diverse parameters and assumptions helped provide recommendations to reduce the required separation distances. These findings are valuable for broadcasters, mobile operators, and regulators in establishing the technical coexistence conditions for DTTB and IMT in the new band.” Is there a proof that shows these parameters were not being considered by the expert in the fields?

Response: Thank you for the critical and helpful insights.

Our manuscript presents a critical examination of electromagnetic compatibility issues arising from the co-channel sharing of DTTB and IMT systems within the new frequency range of 470-694 MHz. While the challenges of coexistence in the previous digital bands (700 MHz and 800 MHz) have been extensively discussed in the literature, our research holds unique significance and novelty.

Our contribution responds to the call from the International Telecommunication Union - Radiocommunication Sector (ITU-R) to conduct comprehensive sharing and compatibility studies, as mandated by Resolution 235 of WRC-15 and the preliminary agenda item 1.5 of the upcoming WRC-2023. What sets our work apart is that, as of now, there is only one published research addressing the evaluation of technical feasibility and requirements for harmonious coexistence in the new frequency band, 470-694 MHz. (This research is cited and referenced as reference [10] in our revised manuscript).

Our contribution is comprehensive, encompassing all potential interference scenarios contingent upon the type of IMT interfering link (Uplink or Downlink) and the DTTB reception type (fixed outdoor or portable indoor) in both rural and urban environments. This comprehensive approach has yielded valuable insights into interference dynamics. Furthermore, we meticulously analyzed various parameters and appropriate mitigation strategies, offering recommendations to reduce the required separation distances. The outcomes of our research hold great importance for spectrum regulators, broadcasters, mobile operators, and policymakers who are grappling with the complexities of spectrum allocation and coexistence between diverse services. The detailed insights and recommendations presented in our manuscript will contribute to making informed decisions and developing effective regulatory frameworks.

In contrast, most previous reference studies in previous digital bands typically considered specific scenarios and use cases. Also, they investigated only the effects of specific parameters.

Point 2. It would be a good author to present a table of comparison of performance with currently used systems in literature and field and compare their performance with the proposed approach in the work and clearly show the difference of performance. The table of comparison can present the advantages and disadvantages of the proposed design approach compared to the currently used approaches.

Response: We would like to express our gratitude for this suggestion.

Investigating coexistence issues between Digital Terrestrial Television Broadcasting (DTTB) and International Mobile Telecommunications (IMT) systems in digital dividend bands involves several approaches, including link budget analysis, simulations, laboratory measurements, and field tests. Each of these methodologies serves a distinct purpose and has its advantages and limitations.

In the revised manuscript (specifically in the introduction from line 81 to line 87), We have included Table 1, which compares the purpose, advantages, and limitations of coexistence investigation approaches.

In this contribution, we have chosen to use simulations, in particular through the specialized software tool SEAMCAT, to investigate coexistence issues between DTTB and IMT systems for several compelling reasons: It offers a cost-effective, safe, controlled, repeatable, and flexible way to assess system interactions, accelerating the analysis process and providing valuable insights for decision-making and system design. Simulations grant complete control over test scenarios, allowing the creation of a wide range, including worst-case scenarios, which may be difficult or impossible to conduct in the real world. This control facilitates comprehensive testing and analysis. Additionally, international spectrum regulatory authorities follow the practice of simulating coexistence scenarios between DTTB and IMT systems, as specified in international agreements.

(We explained why we chose simulation in the revised manuscript from line 186 to line 194).

Point 3.  The literature overview of the work must be improved. There are a total of 24 cited works, which is a very low citation number for such a hot topic. And moreover, there are no cited works from 2023 and only 2 cited works from 2022, and only 3 cited works from 2021.  For such a topic that is being actively persuaded by many researchers, this is a weakness for the work. The ratio of the recently cited works (after 2021) must be at least 25% of the total citation and the total citation for both literature overview and table of comparison must be around 40 or so. Authors are welcome to use the following related works on 3D printable FSS to their literature overview and table of comparison.

Response: Thank you for drawing our attention to improving the literary overview of the work and citing more related works.

In our revised manuscript (specifically in the introduction from line 77 to line 182), we have extensively reviewed the relevant literature according to the approaches used to investigate coexistence issues in digital dividend bands. These approaches encompass link budget analysis, simulations, laboratory measurements, and field tests.

It's worth noting that, as of now, there is only one published research addressing the evaluation of technical feasibility and requirements for harmonious coexistence in the new frequency band, 470-694 MHz. (This research is cited and referenced as reference [10] in our revised manuscript).

We have included Table 1, which compares the purpose, advantages, and limitations of coexistence approaches with reference to the literature works that have employed each approach. Additionally, we made a concerted effort to ensure that the ratio of the recently cited works (after 2020) is as sufficient as possible.

We also cited the two related works on 3D printable FSS which were recently published in the Sensors journal. In the revised manuscript (from line 130 to line 140), We mentioned the two related articles and referred to them in references [25,26].

====================

==============

========

Round 2

Reviewer 1 Report

This paper is well revised. No further comments.

Author Response

We would like to extend our heartfelt appreciation to Reviewer 1 for evaluating our article. Your valuable insights and guidance have greatly improved the quality of our manuscript.

Reviewer 2 Report

I have carefully read the authors response to my questions and the modifications they have performed on the manuscript. In the revised manuscript they reviewed the relevant literature according to the approaches used to investigate coexistence issues, which include link budget analysis, simulations, laboratory measurements, and field tests. They also identified published works that propose field tests to validate simulation results. However, forgot to mention some part of the literature that deal with coexistence issue with field test in the UHF band which include also the specific 470-694 MHz frequency band and practical use of real devices to assess it., as for instance:

Fadda, et al. An unlicensed indoor HDTV multi-vision system in the DTT Bands (2012) IEEE Transactions on Broadcasting, 58 (3), Angueira, P., et al. 

I think this works should be included to complete the literature review.

Author Response

We sincerely thank Reviewer 2 for the constructive suggestions. Your input has significantly contributed to the quality of our manuscript.
Note: The new modifications related to Revision (Round 2) are marked in blue in the revised manuscript.

We are grateful for the recommendation to include the specified reference to enhance our literature review. 
In our revised manuscript (specifically in the introduction from line 167 to line 172), this research is cited and referenced as a reference [40] in the revised manuscript.

Reviewer 3 Report

Authors have made an effort to improve the quality of the proposed work with respect to the reviewer’s comments. The reviewer has no further comment for this work. The work can be accepted as it is.

Author Response

We sincerely thank the Reviewer 3 for thoroughly evaluating our manuscript. Your comments have been instrumental in helping us improve our manuscript.